# Patient and Parent Experiences with Group Telerehabilitation for Child Survivors of Acute Lymphoblastic Leukemia

**DOI:** 10.3390/ijerph18073610

**Published:** 2021-03-31

**Authors:** Genevieve Lambert, Nathalie Alos, Pascal Bernier, Caroline Laverdière, Kenneth Drummond, Noémi Dahan-Oliel, Martin Lemay, Louis-Nicolas Veilleux, Dahlia Kairy

**Affiliations:** 1Department of Experiemental Surgery, McGill University, Montreal, QC H3G 1A4, Canada; genevieve.lambert@mail.mcgill.ca (G.L.); kenneth.drummond@mail.mcgill.ca (K.D.); noemi.dahan@mcgill.ca (N.D.-O.); ln.veilleux@mcgill.ca (L.-N.V.); 2CHU Sainte-Justine Research Center, Montreal, QC H3T 1C5, Canada; nathalie.alos@recherche-ste-justine.qc.ca (N.A.); pascal.bernier.hsj@ssss.gouv.qc.ca (P.B.); caroline.laverdiere@umontreal.ca (C.L.); lemay.martin@uqam.ca (M.L.); 3Shriners Hospital for Children, Montreal, QC H4A 0A9, Canada; 4Shriners Hospital for Children, Université de Montréal, Montreal, QC H2V 2S9, Canada; 5Department of Otolaryngology, Université du Québec à Montréal, Montreal, QC H2X 1L7, Canada; 6Centre for Interdisciplinary Research in Rehabilitation of Greater, Montreal, QC H3S 1M9, Canada

**Keywords:** exercise therapy, rehabilitation, acute lymphoblastic leukemia, intervention study, telehealth, patient perspective

## Abstract

Background: Acute Lymphoblastic Leukemia (ALL) is the most common pediatric cancer. ALL and its treatment cause altered bone-mineral homeostasis, which can contribute to musculoskeletal late adverse effects (LAEs). With the increasing number of childhood cancer survivors, LAEs are reported often, and are aggravated by inactive lifestyles. A telerehabilitation program is proposed to strengthen the muscle–bone complex and prevent future impairment. Objective: This study aimed to explore and better understand patient and parent experience of a telerehabilitation program after completion of ALL treatment. Methods: ALL survivors (*n* = 12), 75% girls, 7.9 to 14.7 years old, within six months to five years of treatment, were recruited to participate in the proposed study, along with a parent. The 16-week group program included 40 potential home-based physical activities, with monthly progression, supervised by a kinesiologist, through an online telerehabilitation platform. Patients could be included in the study if they joined during the first month of intervention of their group (minimum 12 weeks of intervention). A semi-structured post-intervention interview was conducted with the patients and their parent during the final assessment, along with a review of the kinesiologist’s clinical notes, to obtain a portrait of the participants’ experience with the telerehabilitation program. Overarching themes were identified by one author and confirmed by two senior authors before extracting the various aspects of each theme. Results: Of the 12 patients recruited, three were excluded from the analysis because they did not complete the minimum 12 weeks of intervention (one = relapse, one = failure to meet technical requirements, and one = abandoned due to parent’s disinterest). The nine patients who completed the program (six girls; 10.93 ± 2.83 years) had a mean adherence of 89%. The overarching themes identified were the program modalities (group approach with patient–parent paired training, supervised by a kinesiologist), the telerehabilitation system, the participants’ perception of the benefits, and recommendations and suggestions from the families. Both patients and parents expressed very high satisfaction with the program and perceived benefits. Conclusion: Participants appreciated the program and reported they would all recommend it to other families in similar situations. The telerehabilitation method of service delivery was perceived by some as decisive in choosing to participate, while the supervision and intra- and inter-family interactions were the motivating factors that were key to program adherence.

## 1. Introduction

Acute Lymphoblastic Leukemia (ALL) is the most common of pediatric hematological malignancy. The survival rate of patients with ALL currently exceeds 85% [1], which highlights the potential for long-term consequences of the disease. These consequences, known as late adverse effects, include cardiopulmonary impairment and musculoskeletal deficits. Physical activities and exercise programs provide physiological and mechanical stimulation proven to benefit muscle and bone as well as cardiovascular health [2,3]. However, while physical activity and exercise are beneficial to cardiopulmonary and musculoskeletal function in young patients in remission or survivorship, studies show that they have lower levels of physical activity than their healthy counterparts and do not reach the required level of recommended physical activity [4].

Over the years, studies have identified multiple barriers to participation and adherence to exercise programs [5,6,7]. Due to the complexity of management and the rarity of the diagnosis, children affected by ALL often have to attend subspecialized tertiary care centers. Those centers are usually located in large cities and, consequently, not necessarily close to the patient’s home. Travel to the training facility has been shown to be a barrier to adherence and participation. However, a recent study has reported that young patients are more inclined to perform such activities at home, school, or at the gym than at the hospital or rehabilitation clinic [6]. Pain and fatigue may be another barrier to exercise in patients and early survivors. A study of long-term survivors showed that approximately a third of the sample experienced pain requiring the use of analgesics [8]. Studies involving home-based physical activity and exercise training programs for ALL patients in remission or survivorship have reported low recruitment, adherence, and completion rates [9]. These findings suggest that alternative approaches are needed to make home-based exercise programs appealing to children and teenagers in remission and long-term survivors of ALL.

We recently completed a study to evaluate the feasibility of implementing a home-based supervised telerehabilitation program for children and teenagers in ALL remission [10]. Results confirmed a low recruitment rate (21%), as reported previously, which was mostly attributed to the recruitment methodology (i.e., research assistant calling families 2–3 weeks prior to the program, at various time in the survivorship period). To address the recruitment challenges, the previous article proposed integrating the recruitment to patient’s routine medical appointments in early-survivorship. Nonetheless, results also showed that adherence and completion rates were high (89% and 75%, respectively). During the post-intervention evaluations, semi-directed interviews were conducted with patients and parents who completed the program, to better understand the role of supervised telerehabilitation in the high adherence rate reported. This study explored the patient and parent experience of a telerehabilitation program [11] after completion of treatment for ALL.

## 2. Methods

### 2.1. Study Design and Recruitment

This qualitative exploratory study [12] was embedded in a telerehabilitation trial initiated at Sainte-Justine University Health Center, after receiving ethical approval from the Institutional Review Board (2018-1555: e-S@@VIE) in 2018. The trial aimed to recruit a sample of 10 patient–parent pairs to complete a telerehabilitation exercise program after their treatment for ALL. Participants were screened by the hematology oncology service medical team (nurses and physicians) for eligibility. Patients were eligible for the trial if they (1) had a diagnosis of ALL or B lymphoblastic lymphoma, (2) were between 6 and 18 years old, (3) were within six months to five years of treatment completion, and (4) were able to join their group within the first four weeks of the program start for a minimum of 12 weeks of intervention. They were ineligible if they had, as seen in the first publication emerging from this pilot project [10], unresolved fractures or avascular osteonecrosis, a treatment regimen that included bone marrow transplant, or major physical or functional impairment preventing from exercise at the time of recruitment. Technical reasons for exclusion included having an unstable or no Internet connection. Twelve patient–parent pairs were recruited, and nine completed the program, including the initial and final evaluation.

### 2.2. Study Procedures

The Telerehabilitation Program

The telerehabilitation trial was a single-arm interventional pilot study to assess the feasibility of group telerehabilitation programs for early ALL survivors. Patients were contacted to provide information and to screen for interest. Families who were interested were invited for an initial assessment at the hospital. Upon arrival at the hospital, parents of patients under 18 years old provided signed informed consent, and patients between 6 and 17 years old provided written informed assent. After the baseline evaluation of patients’ functional health and musculoskeletal parameters, the kinesiologist visited the families’ home to help them with the technologies and room set-up before starting the 16-week program. Each family was provided with training material and a training watch (Polar A370, © Polar Electro Oy 2020, Polar FlowSync 3.0.0.1337, Finland). A videoconferencing application (Zoom Pro license, Zoom Video Communications, Inc., USA) was installed on the preferred technological tool of each family (tablet, laptop, or desktop computer) [13]; the layout of the videoconferencing interface is shown in Figure 1.

The training program itself was designed to be progressive, to increase either in duration (from 35 to 45 min) or frequency (from two to three times per week) each month. During each session, exercises were adapted or changed according to individual participants’ pain, reported as a number on a scale from zero to 10 (NRS-11) [14], along with a description of the sensation and its location, in addition to its evolution over time and with movement. The patients were reassessed after completing the 16-week intervention. The full study procedures are provided in Reference [10] (Lambert, G., et al. 2021 doi:10.2196/preprints.25569). The study was approved by the Sainte-Justine University Health Center Research Ethics Committee.

### 2.3. Data Collection

As part of the final evaluation, individual semi-structured interviews were conducted with the nine families that completed the study, in French or English, to explore the patients’ and parents’ experience of the telerehabilitation service provided [11]. All interviews were conducted in the same manner, following a semi-structured interview guide, including open and closed questions (see Appendix A for interview questionnaire). The interview guide was adjusted as new information emerged. This led to the addition of a question about the perceived benefits of the program after the second cohort, as the first three families mentioned many perceived benefits throughout the program. Patients and parents were interviewed separately using an in-person format with a research team member. The kinesiologist who had conducted the 16-week intervention program was not present during the interviews to limit participant’s bias in answering. Interviews were audio-recorded, transcribed verbatim, and anonymized during the analysis with the senior authors. Additionally, notes from the kinesiologist supervising the program were reviewed for relevant content reporting participants’ feedback regarding the service or system, in addition to their barriers to interventions.

### 2.4. Data Analysis

The first author (G.L.) identified major overarching themes, using the interview transcripts. The themes were then discussed with two of the senior authors (K.D. and L.N.V.), to confirm that the themes reflected and transcended the interviews. A second read-through of the interviews was done, using the Taguette free application for qualitative analysis (https://app.taguette.org, accessed on 19 November 2020) to identify quotes representing different aspects of the themes, and the number of participants (parents or patients) mentioning them. The results are presented according to overarching themes, reporting similarities, and divergences between patients’ and parents’ experience of the program. Quotes selected for reporting from the French interviews were translated to English by a bilingual author (G.L.).

To complement the perspective captured in the interviews, a list of barriers encountered throughout the program by participants was compiled. The kinesiologist’s clinical notes were reviewed to identify two principal types of barriers: (1) technological challenges and (2) pain reported by either the participant or parents. Since pain varied in location, intensity and duration, we reported the number of sessions where adaptations were required due to pain relative to the total number of sessions completed. Technical challenges were considered major for the Zoom videoconferencing system if it led to discontinuation of the session, and minor if it hampered communication between the families and the kinesiologist (e.g., video or audio lag). Technical challenges for the training watch were coded as major if no data were acquired (e.g., uncharged watch or patient forgetting to put it on, resulting in an absent HR monitor and step count), and minor if there were difficulties charging the watch, low battery, or the participant forgot to start the watch at the beginning of the session, resulting in inconsistent data collection (incomplete or absent HR monitor).

## 3. Results

### 3.1. Participant Characteristics

Of the 12 patient–parent pairs recruited for the telerehabilitation program, nine from four different groups completed the program and final interview. The three families that did not complete the program were either excluded (*n* = 1 relapsed; *n* = 1 did not meet technical requirements) or dropped out due to parents’ lack of interest (*n* = 1). The characteristics of the 16 participants who completed the program (parents and children) are found in Table 1.

### 3.2. Overarching Themes

The interview results confirm that patient and parent experiences were influenced by the modalities of the program (i.e., group training, patient–parent pairing, and kinesiologist supervision), the perceived benefits of the intervention, and the telerehabilitation system itself. Participants also offered recommendations for other families and healthcare professionals considering such a program. The themes identified and confirmed by the authors are shown in Figure 2.

### 3.3. The Group Approach

The role of the group approach in participants’ experience was widely positive. Participants reported few challenges. Only one parent reported a major challenge.

While only some parents mentioned the telerehabilitation program would have been impossible without the group approach, many patients mentioned that when the other families were unable to attend the session, they felt it was “a little boring” (Patient 06) or “monotonous” (Patient 12), suggesting it was less motivating. In fact, the group approach was identified as a motivating factor by all participants, parents and patients alike. The underlying motivational phenomena were mostly healthy competition and a collaborative atmosphere between families, in addition to feeling able to relate to each other. Patients reported “healthy competition, because when you’re next to someone, you’re always going to try to do like the person next to you (during in-person trainings). But now, it is in our home… (everyone feels comfortable) That’s it, everyone’s doing their work. When someone feels tired, others can go on, or stop, you don’t feel like you have to force or anything” (Parent 12). The second motivating phenomenon of the group approach was the collaborative atmosphere: “It was like teamwork” (Patient 09). It was reported that if “someone had a problem, we could give ideas, we could help each other” (Patient 11) and encourage each other: “It’s actually more than positive, because when (Patient 09) would see the other girl exercising, she would be like, ‘*Yeah, let’s do this’!* and they would high-five each other at the end” (Parent 09). Moreover, participants appreciated being able to relate to others about their health: “I wasn’t comparing myself, but I could see that I wasn’t the only one who wasn’t used to it. So that made me feel comfortable, I was not alone” (Parent 05). “There was a father who had a knee problem, and I have a shoulder problem” (Parent 12). “It was good because you could meet more people and could see others who had the disease, what it is like, and if they have the same booboos as you do, things like that” (Patient 04). They also mentioned shared challenges: “Because sometimes they’d say things to each other. Let’s say, if it’s advice, it can also apply to me too. Even if she (the kinesiologist, G.L.) isn’t talking to me, I can still listen to her” (Patient 10). “We are not alone in our living room doing this” (Parent 10).

The level of interaction varied between groups. Group 1 did not communicate very much with other families, while Groups 3 and 4 had patients that played together when hospitalized during treatments. “I like exercising with someone—another family I’ve met in the hospital. Before, it was kind of a hospital friend” (Patient 09). This may have elicited a higher level of interaction among both patients and parents. Participants in Group 4 did not know each other before the program, but became friends: “(today we) met for the first time (during the final evaluation), it’s like (we have) known each other for years” (Parent 12).

During the interviews, very few challenges were mentioned related to the group approach. Parents commented that families “were not always in sync” (Parent 10), which made the parent anxious to keep pace, although she added that the kinesiologist did not pressure them. Another parent mentioned not really hearing the other families “people were still discreet” (Parent 04). On the other hand, challenges regarding the group approach reported by patients included families disappearing from the screen, or not seeing everyone all the time; however, only one patient said this approach was distracting. Another patient reported that, at times, another participant looked like he did not want to be there.

The major challenge cited earlier referred to a parent experiencing discomfort overhearing an incident in another family: “We didn’t have to hear that” (Parent 11). The parent said, “We don’t all have the same values”, and concluded, “Afterwards, I don’t know if someone spoke to them or not, but it was less disturbing (the training), and more fun”.

### 3.4. The Patient–Parent Paired Training Experience

According to the parents, the patient–parent paired training was also beneficial. Limited challenges and no major challenges were cited by study participants.

The main benefits reported were the motivation it provided, the time spent together, the healthy competition, and the opportunity for parents to help their child. Motivation was mentioned by several participants: “It also helps motivate (Patient 06) because sometimes she doesn’t want to do it, sometimes she does. It depends on her mood” (Parent 06). Many patients or parents mentioned that training sessions were an opportunity to spend quality time together: “It’s rare that you have a lot of time together because, at school, you always have to hurry. When it comes to training, you are more relaxed” (Patient 04). Many parents found it to be a time to develop their relationship with their child: “We do it together. The exercises with the elastic band also created a bond because we watch each other and do it together. I think that it gave me a (sense of) complicity with my daughter (a desire) to do activities (together), and it really makes me see that training should not be seen as an overload of activity, but rather as a family moment” (Parent 05). One of the parents reported that his relationship with his son improved. The child “has ADHD with an opposition disorder, so there’s always parent–child conflict. (The program) allowed me (Parent 12) to spend time with him, and at the same time, to do something with him, (…) to get closer—well not to get closer, but (for him) to see someone other than the father who says stop doing that, or whatever” (Parent 12). Both parents and patients also appreciated that there was healthy competitiveness between them. This rivalry was perceived as motivating: If patients saw they were able to do one more repetition than their parent, it empowered them to continue. Moreover, patients appreciated the opportunity for their parent to assist them: “It helped (to do the training with my mother) because my mother, even if (the kinesiologist) didn’t say it, she (referring to the mother) could tell me things that I had to improve” (Patient 05). Parents recognized and valued the supportive role they played in their child’s experience: “(…) him just doing the exercises wouldn’t have worked. First, (he has to understand), and also there are exercises you (referring to Patient 07) needed a little help with and stuff—like holding the carrot (referring to assisted drop jump)—it went really well. And then, I also participated, and that makes me feel good, too” (Parent 07).

As indicated above, the pairing of parents and patients caused no critical issues; however, some challenges were raised, such as conflicts: “sometimes, if we quarrelled (…) everyone could hear us” (Patient 04). One family mentioned that a parent was less motivated, which reduced the motivation of the patient. Another child mentioned that sometimes the parent was anxious to finish quickly to continue doing housework. Differences in strength between partners was also reported twice as a possible limiting factor by two teenagers, which had an impact on the partner they chose to exercise with: One decided to exercise with her parents instead of her siblings; another chose his father instead of his mother as his exercise partner.

### 3.5. The Training Experience While Supervised by A Kinesiologist

When asked about their experience of training while supervised by a kinesiologist, participants unanimously said they felt the kinesiologist had a positive impact. All participants said that they received enough information to do the exercises correctly, and that the information was clear. Some mentioned that, if they misunderstood or an audio-video lag occurred, the kinesiologist took the time to explain and demonstrate the exercise again. This was greatly appreciated by participants, as expressed by one parent: “If (…) we didn’t understand, she would go down on the ground to demonstrate the exercise” (Parent 03). Participants expanded mainly on three elements regarding the kinesiologist: (a) the kinesiologist’s personality, which helped foster the therapeutic relationship; (b) their understanding and attention paid to pain and discomfort; and (c) their knowledge and ability to adapt the training to individual limitations.

The kinesiologist’s personality was the factor most often mentioned by study participants. Many parents used qualifiers such as attentive, knowledgeable, motivating and motivated, inclusive, accommodating, accessible or approachable, and open to questions. Additionally, almost all parents used at least one of the following adjectives, warm, kind, playful, positive, dynamic, or enthusiastic, when describing the kinesiologist’s personality. For their part, patients described the kinesiologist as friendly, nice, happy, funny, playful, motivating, and engaged. Patients appreciated the incorporation of games in the training “We do games, yes it’s going to be exercise, but in activities” (Patient 06). Some patients mentioned that the kinesiologist was a mix of serious and comical, or relaxed, that they were supportive, and that they provided constructive criticism. To describe their relationship with the kinesiologist, some made the following comparisons: “like my teacher” (Patient 11) or “a fourth cousin who you don’t know, but play well with” (Patient 12), illustrating the friendly therapeutic relationship that developed between the patient and trainer.

Furthermore, some parents and patients emphasized that they appreciated the opportunity to understand the nuance between pain and discomfort that the patients were experiencing. As seen in Table 2, patients frequently reported pain throughout the program which demanded adaptation (varying among patients from 42% to 96% of sessions). The kinesiologist took the time in each instance to discuss and understand the pain or normal discomfort that the participant was feeling. On this topic, one parent interviewed said that, before the program, he did not know how to react when his son was complaining of pain, but he was now able to ask questions and help his son understand what he was experiencing: “Yes, he had pain in his feet, because it is normal after treatments. But now, we learned something: *‘is it pain or discomfort’?* Sometimes, he does physical education at school, then he comes home, and he’s tired. And ‘*yes, you have done physical activities, so, of course, you are a little uncomfortable*’. (…) Then he understood the difference” (Parent 07).

Lastly, the kinesiologist paid careful attention to pain, in order to adapt the exercises and training to participants’ individual condition and limitations. The ability to adapt the exercises and training to all participants was greatly appreciated by both parents and patients. “And really, it worked” (Patient 11), “I had back pain before, and she gave me tips to have less back pain” (Patient 04). Parents appreciated the adjustments for their children as it made the training “adapted to their needs” (Parent 04). But they also appreciated having access to the service for themselves: “because, even though we are parents, we have little booboos” (Parent 12), and “she always had a plan B and a plan C when someone was having pain; you know, like me. Sometimes I had pain in my knee, and I would tell her and (she would say:) *Okay, do this exercise instead*” (Parent 06). This allowed parents “to be active, without doing the same exercises” (Parent 12).

### 3.6. The Perception of Training Benefits

Although the program aimed to improve physical function, participants perceived benefits that extended beyond this outcome. The interviews and kinesiologist’s clinical notes revealed that participants recognized positive effects on emotional or social health, and changes in appearance and function. Some participants also reported less pain and discomfort as a result of the program.

The effects on emotional or social health were reported mainly by the parents on behalf of their children. Almost half the parents spoke of the effects of the program on their child’s self-esteem, ability to focus, general motivation, or socialization and communication skills. Two patients also mentioned this aspect during their interviews:

Patient 09, “It also helped me get less distracted”.

Interviewer, “Less distracted? Do you mean during the training, or just in general”?

Patient 09, “In general”.

Interviewer, “In general”?

Patient 09, “It helped me get less distracted, because I’m always distracted”.

The main changes in physical function cited by parents and patients alike were improved strength, increased endurance either during trainings or in everyday life, and increased energy or less fatigue. Participants mentioned, among other things, the ability to do exercises at the end of the program that they could not do at the beginning (such as push-ups or burpees), or being able to wear their backpack to school, walk to school, jog in a corridor, or pick up grocery bags more easily than before, with less tiredness or fatigue. “At first, he was really, in physical terms, he was really not at all in shape. I even met his physical education teacher at school, and he said, ‘*Ah! (Patient 07) has improved. He is doing very well! Are you doing anything special*’? I explained the program to him a bit. We were also able to start playing other sports, and he is good at them: We play badminton and soccer, and it’s going well. Before, I tell you, five minutes and he was ready to give up” (Parent 07). Another patient decided to take up a new sport: “Because I wasn’t there for a few years. Just after I finished chemo, and I wasn’t able… I was weak… I didn’t have enough energy, strength so—I felt like, during the training I did, I bounced back, so I want to take advantage of it to try to start karate again” (Patient 06). Parents also spoke of appearance. A third of parents reported changes in their child’s appearance, mainly patients looking thinner or having more muscle definition. One parent also reported that they, too, lost weight.

Some patients and parents saw a reduction in pain and discomfort as a result of the program. Two parents expressed a reduction in either chronic or acute knee pain due to the program exercises or movement corrections. Patients mentioned chronic pain such as headaches, and back, leg or foot pain, which they attributed to the lasting effects of ALL treatments. Patients did not report diminished frequency of headaches in daily life, however, some said that other types of pain decreased or disappeared: “All the pain went away while doing the exercises. I may still be in pain sometimes (referring to headaches), but the physical pain, doesn’t happen that much anymore” (Patient 10). Another patient reported disappearance of pain multiple times during the program: “I used to get up (in the morning), and there was a leg like, it just didn’t work. It was stiff, and it hurt. I had to find a (different) way to go down the stairs, or it hurt. But now, that has stopped” (Patient 11). Clinical notes confirmed that some patients reported specific sensations as pain at the beginning, and over the weeks reported it as discomfort or no longer reported it at all. This was most notably the case for Patient 07 and Patient 12.

### 3.7. The Telerehabilitation System

One of the main benefits of telerehabilitation technologies mentioned by the parents was that the system was adaptable. The families valued being able to use their own technologies. For example, a family that used their tablet mentioned, “it wasn’t distracting me from anything and, you know, if it had been a big screen, and I would have seen them (the other families) live, then I would have been uncomfortable. But the way it was set up, it was fine” (Parent 03). Another family said, “We really had a great set-up at home. I was able to take my laptop and put it (the video feed) on the television” (Parent 05). Furthermore, many patients and parents described the system technologies as user-friendly, which allowed patients in more than half the families to sometimes connect by themselves to the telerehabilitation sessions. A specific detail mentioned that facilitated the experience was that links for the training sessions were sent by email the day before the session: “It was easy. Well, we just had to click on a link (…) in an email. Because we had the codes, and we were given them as we went through the sessions” (Patient 12).

However, even if the technologies were generally easy to use for all families, when asked about minor or major challenges, most participants mentioned occasional video or audio malfunctions, “It happened from time to time, sporadically” (Parent 12). System difficulties were reported if other family members were using the Internet simultaneously, if the weather was poor, or sometimes if families were doing the training from somewhere other than at home. Some families said that issues occurred mostly in the early sessions, while they were adjusting their Internet consumption: “At home, the children all have devices that use the Internet. So, at the beginning, we weren’t sure. We didn’t really know how much consumption it was going to take. So, at the beginning, the first and second week we had a lot of, you know, lags, because the Internet was in great demand, because the children were already listening to videos with voices, and we had the camera live with the voice. Well, there was one point when the Internet jumped a bit there. (…) We had stopped the devices (for the other) children during the training; to dedicate the Internet just to that” (Parent 06). “No, nothing went wrong. But at first the camera froze a lot”. (Patient 09). Two other factors were mentioned as limiting by some parents. The first was the sound, “As soon as someone spoke, we couldn’t hear what was happening elsewhere (…) we could no longer hear what the instructor was saying. It was hard to manage sometimes. It was not all the time” (Parent 10). The second limiting factor was the watch, “Sometimes we forgot, it’s our fault, but we forgot to plug in the watch, and things like that” (Parent 04).

Overall, the telerehabilitation method of delivering the service was identified as beneficial by all participants, because the program did not require travel and could be done from the comfort of their home, allowing families to balance training with personal and professional activities. Some mentioned that it might not have been possible for them to participate if the program had been in-person: “We don’t waste 20 or 30 min to get there and come back. It sounds like nothing, but it’s still precious. Because if we had to go to (the hospital), (…) it’s not certain that we could have participated” (Parent 06). Others said that the telerehabilitation delivery was crucial in their decision to participate “Three evenings, it was going to be a little more difficult. (…) The fact that I knew I didn’t have to go out, that was one of the characteristics that made me say yes” (Parent 11). Patients also liked not having to come to the hospital for interventions: “Before the telehealth, I was like ‘*Why don’t we go to the hospital? That would be better*’. It would be weird to do it on the computer. But then I noticed it was better, because you can do it in your living room. You don’t have to be like, ‘*Okay, it’s 6:30 a.m. I have to go to the hospital*’. Because it takes an hour to get there, and it will be at 7:30 a.m. You can say like, ‘*It’s 7:15. Okay, I’m going to turn on the television, and I’m going to start this*’. There you go, you didn’t have to go to the hospital and do everything” (Patient 05). Doing the training at home was appreciated by all parents, but this subject was not addressed as much by patients. Parents mentioned that their children felt at ease doing their exercises, and that it was even more beneficial in winter that they did not need to leave the house. Moreover, they agreed on the convenience of the delivery method. Most parents mentioned that it helped to balance the training in the family’s everyday life: “In family life with children, it fits well, it makes it much easier. I don’t need to call a babysitter for the others while we go to his appointment. For family management, I think it’s something very useful” (Parent 10). As indicated in the kinesiologist’s notes, it was common for families to have the other parent involved in the training (*n* = 7), as well as siblings (*n* = 4), cousins (*n* = 1), grandparents (*n* = 2) or even friends (*n* = 2).

However, in some cases, the increased accessibility of the telerehabilitation method of delivery led to a sense of obligation, especially among the teenagers: “(I) felt a bit like I had to do it even though I didn’t feel like it sometimes. Then, like when I was in pain, I felt bad for not doing it” (Patient 03).

### 3.8. Recommendations and Suggestions from the Families

When asked if they would recommend the program to other patients and their families, all parents and patients agreed that they would, some even going so far as to propose that it become part of the ALL treatment protocols: “This should be a part of the treatment. It has (been) five years (since) she (Patient 09) got sick, she has (had) two years of chemotherapy. This type of program should be included in her treatment, as a follow-up with the hospital, because she would benefit a lot. (It would be) beneficial for all kids in her situation” (Parent 09). Similarly, most parents and patients also thought that the telerehabilitation delivery method could be useful not only for cancer survivors but for anyone interested in exercising. Some patients specifically mentioned that it would be ideal for populations with factors that limit travel or mobility. Others mentioned that if the person is able to do the training in person, it may not be needed.

Many parents agreed that they would propose the telerehabilitation method of delivery to another health or sport professional if they ever needed such rehabilitation services again, mainly due to the aforementioned general benefits of the telerehabilitation service delivery method: no travel time, being able to do it from the comfort of their own home, and the balance of training with personal and work life.

During the program, a few families said that they would have appreciated at least a few in-person training sessions to help correct movements that were more technically difficult for the patient and to get tactile feedback. Other suggestions from the families to the research team for improving the system or service were as follows:Provide more material (such as yoga mats, or wider variety of weighted balls, and elastic bands).Provide the system to families with fewer technologies at home, or with better quality cameras that could be controlled by the kinesiologist to focus on participants so that they would stay in the frame while in motion (Parent 06), or even to use a set of three cameras to see different angles.For the kinesiologist to “have (real-time) access to the data from the training (to) watch” on the screen, to help motivate and set goals (Parent 10).Have a more flexible schedule for families to do a different session if they couldn’t attend one earlier in the week. “Let us decide when our training is. (…) Sometimes, we might have an exam on Thursday, so we can’t do it on Wednesday evening because it takes too long and we couldn’t study” (Patient 04), or for families to select their preferred times without selecting the age group.Offer a longer program to gain more benefits (“a full school year” (Patient 11) was proposed by two families).Incorporate a wider variety of exercises, or exercises that each child likes, and more games.Encourage families to include friends, siblings, and other family members to shift the focus and make it playful.Offer the service earlier to help families plan their schedules ahead of time.Offer the service to more families “As a part, included in the treatments” (Parent 09).The hospital should better advertise the service to families, such as having the treating physician recommend the program to patients during the final months of treatments.

## 4. Discussion

Parents and patients were very satisfied with the program, as confirmed by the semi-structured interviews. All participants said they would recommend the program to other families in their situation. In addition, they all spoke in great detail of the benefits of the program and its modalities, and were less concerned by its challenges.

Factors that contributed to the high satisfaction identified in this study, i.e., primarily the method of delivery and the intervention modalities, may explain the high adherence rates recorded in this study (89% [10]), and for some, may have contributed to the initial decision to participate. The current findings illustrate that the telerehabilitation method of delivering the rehabilitation service was critical in choosing to participate, played an important role in facilitating participants’ access to the program, and, subsequently, to their satisfaction with the program. Pairing, grouping, and supervision also positively influenced participants’ experience once they chose to enroll in the program. The frequency and quantity of benefits mentioned during the interviews concerning these aspects of the program far outweighed the challenges. Therefore, it would seem that the modalities of the telerehabilitation service (i.e., pairing, grouping, and supervision) impacted adherence to the program, while its accessibility contributed to the initial decision to participate.

The literature examining factors influencing ALL survivors to participate in, adhere to, and complete a rehabilitation or exercise program is limited. Nevertheless, the findings of the current study are consistent with the earlier study done by Wright et al. (2015). Both highlight that participants prefer to exercise at home rather than at a hospital or clinic [6].

Furthermore, the current study parallels the findings of Kairy et al. (2013) in certain respects. The latter explored the patient perspective of adults who received an eight-week home-based telerehabilitation program following total knee arthroplasty [11]. That study used a system similar to the current study, to provide supervised intervention to patients; however, their program was neither paired nor grouped. Although each study targeted very different populations, their findings are astonishingly similar, indicating that the aspects interviewees appreciated most were being able to receive the services at home, and their relationship with the physiotherapist. In both studies, participants reported that the physiotherapist and kinesiologist were capable of adequately evaluating patient’s technique, pain, and discomfort. Furthermore, patients in both studies mentioned that some in-person sessions might be useful to facilitate a clinical follow-up, such as physical correction. Lastly, in both studies, telerehabilitation was not viewed by patients as an impediment to their satisfaction with the program. However, the differences between the populations and modalities (i.e., paired and grouped trainings) may have influenced the perspective of the participants: Patients in the current study highlighted the playfulness of both the training regimen and their kinesiologist. Additionally, while in the study by Kairy et al. the patients preferred the telerehabilitation delivery method over in-person, due to saved preparation and travel time, they did not mention the advantages of better balancing of personal, family and work life, a factor raised repeatedly in the current study. The increased social interactions, due to the paired and grouped approach of the current study, may have imposed some challenges, but motivated participants to adhere to the program [15].

To this extent, a study revealed that children with disabilities considered their participation in physiotherapies and occupational therapies meaningful when they were enjoying themselves, felt capable and autonomous in their activity, and when there was a social component to the activity [16]. The modalities of the present study had an intrinsic social component. Participants took full advantage of the social involvement. The original “patient–parent paired” experience very often expanded to not just two but three participants, including other family members or friends. Furthermore, participants mentioned the perceived benefit of being able to do exercises or tasks that had once been difficult. This finding aligns with the essential role of capacity in creating a meaningful rehabilitation experience. Lastly, participants highlighted the importance of enjoyment in their experience of the current program. This enjoyment is noted, in their perspective, concerning the modalities: Patients appreciated the healthy competition between families, as well as with their parents. Enjoyment is also observed in their perspective of the kinesiologist’s personality. Both patients and parents described her as playful. It is also noted in their recommendations for future programs.

Early survivors of ALL are known to be a difficult population to recruit for exercise interventional study. Their adherence to such programs tends to be low, since they are less active than the general population. Furthermore, after ALL treatment completion, patients and their families may experience challenges related to family cohesion [17] and the addition of school-related activities to the schedule [18]. In these respects, the modalities of the current program were beneficial, since they reduced the burden on the family’s schedule by increasing the accessibility of the rehabilitation services. The social intra- and inter-family interactions might also provide a valuable tool to improve patient motivation to adhere to such rehabilitation programs.

### Study Limitations

The main limitation of this study was the small sample size: Results from nine parent–patient pairs cannot be generalized to the entire population of early survivors of ALL. Furthermore, interviews were conducted only with participants who completed the program, and their perspectives may differ from participants excluded prior to intervention completion.

The current study used a hybrid approach (i.e., not all interactions with the kinesiologist were done remotely) that may have allowed the therapeutic relationship with the participants to develop, but it is impossible to infer whether this would be better than other telerehabilitation approaches, such as conducting all activities online. One of the main aspects appreciated by both parents and patients was supervision by the kinesiologist during the telerehabilitation training sessions. However, it is important to note that the therapeutic relationship fostered between participants and the kinesiologist was not created exclusively through the use of telerehabilitation. Participants met the kinesiologist twice before starting the program: once for the baseline evaluation in the hospital (three to four hours), and once for a home visit to help the families prepare the technologies and their training environment (45 to 90 min). Consequently, it is not possible to infer from the results of this study alone that a therapeutic relationship can be created through the use of telerehabilitation alone.

Moreover, regarding the benefits perceived by participants, it remains unclear if the decreased pain that patients reported was due to actual changes in the pain sensation (i.e., whether it decreased to discomfort or the sensation was no longer present), or was the result of the educational role of the kinesiologist in gaining a better understanding of the pain. Nonetheless, pain is a very important issue and was raised as limiting in this study. This finding raises the need to address pain in this population, using a multidisciplinary approach.

## 5. Conclusions

### 5.1. Summary

The telerehabilitation program with early survivors of ALL yielded very high satisfaction among participants. The telerehabilitation method of delivery may have been a key factor in their consent to participate in the project, but the group approach, patient–parent paired training, and supervision by a kinesiologist were the main factors that spurred participants to adhere to the program throughout the 16 weeks of interventions.

### 5.2. Future Avenues

Future studies should further explore the impact of personal and family factors on adherence to telerehabilitation, in order to address challenges and promote participation in such programs. In a similar vein, the reasons why some ALL survivors chose not to participate or abandon during the course of the intervention should be explored, in order to provide solutions and programs that are accessible and appealing to a larger group of patients and survivors. Lastly, future interventional research should focus on assessing the impact of the program on the emotional and social well-being, in addition to quality of life (QoL) and health-related QoL, of participants using validated tools, and should also examine other formats of telerehabilitation interventions, compared to in-person services.

## Figures and Tables

**Figure 1 ijerph-18-03610-f001:**
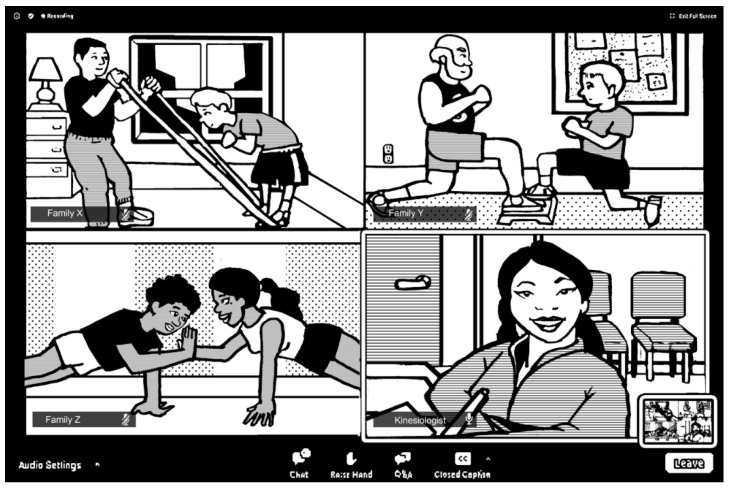
Illustration of videoconferencing interface during a group session, by Nizar Laarais.

**Figure 2 ijerph-18-03610-f002:**
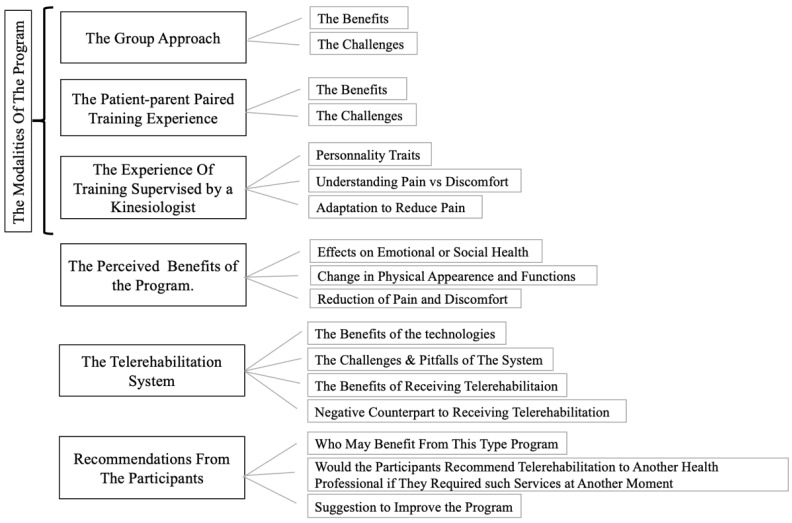
Overarching themes.

**Table 1 ijerph-18-03610-t001:** Participant characteristics and technologies used for the program.

Patients’ Characteristic					Parents’ Characteristic			Families’ Characteristic	
ID	Sex	Age ^a^	Diagnosis and Prognosis	Time since TC	ID	Sex	Age ^a^	Group No.	Technology Used
Patient 03	F	14	ALL; HR	14	Parent 03	F	51	1	Tablet
Patient 04	F	9	ALL; SR	27	Parent 04	F	41	1	Laptop computer connected to television
Patient 05	F	9	ALL; SR	53	Parent 05	F	40	1	Laptop computer connected to television
Patient 06	F	13	ALL; SR	44	Parent 06	M	44	2	Desktop computer connected to television
Patient 07	M	8	ALL; VHR	13	Parent 07	M	43	3 ^b^	Laptop computer
Patient 09	F	8	ALL; SR	34	Parent 09	F	33	4	Laptop computer
Patient 10	F	9	ALL; SR	36	Parent 10	F	36	4	Laptop computer connected to television
Patient 11	M	13	B-LL; SR	57	Parent 11	F	52	2	Laptop computer
Patient 12	M	15	ALL; HR	52	Parent 12	M	44	2	Tablet

^a^ Age reported at the final interview. ^b^ The family in Group 3 is presented alone in this table because they started the program with the family that abandoned mid-program, then joined Group 4 but had to finish the last month alone, due to schedule restrictions. ALL, Acute Lymphoblastic Leukemia; VHR, very high risk of relapse; HR, High Risk of Relapse; SR, Standard Risk of Relapse; TC, Treatment Completion (month); B-LL, B-cell Lymphoblastic Lymphoma.

**Table 2 ijerph-18-03610-t002:** Barriers to interventions reported in the kinesiologist’s clinical notes for each family.

Families	Patients’ Pain (Sessions with Adaptation Due to Pain/Total Sessions Completed)	Parents’ Pain (Sessions with Adaptation Due to Pain/Total Sessions Completed)	Level of Technological Challenges: Zoom		Level of Technological Challenges: Polar Watch	
			Major	Minor	Major	Minor
03	21/30	2/30	2	1	1	1
04	27/28	1/28	1	7	4	3
05	18/39	1/28	-	2	-	1
06	16/37	9/37	-	4	2	4
07	25/38	0/38	3	-	-	-
09	16/39	1/39	-	4	-	2
10	14/33	1/33	-	-	-	2
11	15/31	15/31	-	4	-	1
12	16/25	9/25	1	2	-	-
Total	168/300	39	7	24	7	14

## Data Availability

All data supporting results can be found coded at the Center of Motion Analysis of the Shriners Hopital for Children–Canada.

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
