# Peer review of "Patient and Parent Experiences with Group Telerehabilitation for Child Survivors of Acute Lymphoblastic Leukemia"

_ijerph, 2021, doi:10.3390/ijerph18073610_

Round 1
Reviewer 1 Report
- more authors than patients? are you sure, all co-authors took part in most parts of the research???
- exclusion criteria: physical or functional impairment at the time of recruitment (? so why they need tele-rehabilitation?)
- results section: too long- may be some tables should occur?
- discussion, specially from the "study limitation" sub-title is unclearly written.
Author Response
Thank you for your time, effort, and feedback in reviewing our manuscript! You can find below the respective answers to the corrections you proposed:
- We totally agree with the reviewer that it may seem odd to include more authors than patients. However, the project originates from a grant submission aimed to favor collaboration between institutions. Therefore, six researchers contributed to the design of the study. Further, pediatric cancer survivors are a population requiring complex care, which prompted the need for the clinical team to be involved. The details of the tasks executed by the authors can be found in the initial submission.
- The exclusion criterion you mentioned has been clarified in the text. Major physical or functional impairments (such as amputation, trisomy, deafness, etc.) were considered for exclusion, due to the primary nature of the trial to assess feasibility to deliver a grouped approach program with plyometric exercises. Nonetheless, multiple patients included in the current study had chronic comorbid musculoskeletal (n= 5), metabolic (n=2), and psychosocial (n=3) conditions. In a second step, future studies should elaborate on clinical tools for telerehabilitation systems to be inclusive.
- A lot of the richness of the results is in the verbatim and the way ideas are elaborated. With this in mind, in addition to both the limited literature on the subject and the fact that length was not a concern of the other reviewer or the editor, the authors decided to keep the results as-is.
- In response to the reviewer’s observation relating to the writing quality, the whole manuscript, including the discussion, was revised by a third-party reviewer to ensure that the text would be properly written. Thank you for highlighting this.
Reviewer 2 Report
The authors evaluated the experience of children as survivors of acute lymphoblastic leukemia (ALL) and their parents with an online physiotherapy program. They found that these efforts are useful and were highly appreciated by both patients and parents.
Minor Suggestions for Improving the Manuscript:
- Line 37: Is it known why the program was abandoned?
- Line 80: What would be the definition here of „youth“?
- Line 81: Why was the recruitment rate low? Maybe just enumerate some key points of known reasons – plus possibly reference(s).
- Throughout the manuscript: „Telerehabilitation“ as a word does not please the eye and is not as easy to read as „tele-rehabilitation“ (like in the title of the paper).
- Table 1: This table would benefit from a different format, e.g. maybe the data for each family (patient/parent) in one row; the row below with the next patient-parent pair maybe in a different color or some grey shading, third row again in white and so on.
- Table 1: For the age of the children the position after the decimal point does not appear to be necessary (for example 14 instead of 14.5).
- Legend to Table 1: The abbreviations in alphabetical order so one does not need to go through the whole collection in search of one definition.
- Throughout the manuscript: Minor typing mistakes can be easily corrected, e.g. acessiiblty, intrinsect, suvivors, etc.. Only careful reading and editing necessary.
- Outlook: The authors already mention „future studies …“ in lines 581-583 and in lines 614-616. These could be merged in its own paragraph after „Conclusions“ and expanded a little bit. Maybe as header: „Future Studies“ or „Outlook“.
- Discussion: Maybe the authors wish to also introduce and discuss the term „quality of life“ which has become an important aspect of cancer treatment in recent decades.
Author Response
Thank you for your time, efforts, and feedback in reviewing our manuscript! Here are the responses to the specific suggestions you made:
- Line 37: Is it known why the program was abandoned?
Response: The family that abandoned the program was due to the parent’s disinterest in the program. This information was added in the abstract and can be found in the results section « participants’ characteristics ».
- Line 80: What would be the definition here of „youth“?
Response: The word « youth » was replaced by « children and teenagers » in the text.
- Line 81: Why was the recruitment rate low? Maybe just enumerate some key points of known reasons – plus possibly reference(s).
Response: A one- or two-sentence summary of the challenges and suggestions to improve recruitment proposed in the previous paper was included in the text.
- Throughout the manuscript: „Telerehabilitation“ as a word does not please the eye and is not as easy to read as „tele-rehabilitation“ (like in the title of the paper).
Response: As « Telerehabilitation » is the main term used in the literature and simplifies search strategies to the same extent as « telemedicine » or « telehealth », the term without the dash was selected and used throughout the manuscript.
- Table 1: This table would benefit from a different format, e.g. maybe the data for each family (patient/parent) in one row; the row below with the next patient-parent pair maybe in a different color or some grey shading, third row again in white and so on.
Response: Table 1 has been updated to have patients’ and parents’ information on the same row, and shading has been added to every other row in order to improve readability.
- Table 1: For the age of the children the position after the decimal point does not appear to be necessary (for example 14 instead of 14.5).
Response: Decimals for the age of the children have been removed.
- Legend to Table 1: The abbreviations in alphabetical order so one does not need to go through the whole collection in search of one definition.
Response: We reorganized the legend of table 1 to put the information in alphabetical order.
- Throughout the manuscript: Minor typing mistakes can be easily corrected, e.g. acessiiblty, intrinsect, suvivors, etc... Only careful reading and editing necessary.
Response: In response to the reviewer’s observation relating to the syntax, grammar, and orthography mistakes, the whole manuscript was revised by a third-party reviewer to ensure that the text would be properly written. Thank you for highlighting this.
- Outlook: The authors already mention „future studies …“ in lines 581-583 and in lines 614-616. These could be merged in its own paragraph after „Conclusions“ and expanded a little bit. Maybe as header: „Future Studies“ or „Outlook“.
Response: A subheading was added in the conclusion « future avenues » to merge all the recommendations into one paragraph, as you recommended.
- Discussion: Maybe the authors wish to also introduce and discuss the term „quality of life“ which has become an important aspect of cancer treatment in recent decades.
Response: The aim of the study was to explore participant experience with the telerehabilitation program. Further, as mentioned in the methodology, the health and social benefits were not originally part of the questionnaire. Knowing that quality of life was neither measured formally (HR-QoL) nor was it an intended theme to address, the authors feel that discussing it may be misleading. Nonetheless, it was included in the « future avenues » section for future studies to elaborate on the method of delivering rehabilitation services and their impact on QoL and Hr-QoL of patients and their families.